# Explainable Weakly-Supervised Cell Nuclei Segmentation by Canonical Shape Learning and Transformation

**Pedro Costa**[1,2]                                                                  UP201000588@FE.UP.PT
[1] *Faculty of Engineering, University of Porto*
[2] *Institute for Systems and Computer Engineering, Technology and Science, INESC-TEC*

**Alex Gaudio**[1,2,3]                                                                AGAUDIO@ANDREW.CMU.EDU
[3] *Carnegie Mellon University*

**Aurélio Campilho**[1,2]                                                             CAMPILHO@FE.UP.PT
**Jaime S. Cardoso**[1,2]                                                             JAIME.CARDOSO@FE.UP.PT

**Editors:** Under Review for MIDL 2022

## Abstract

Microscopy images have been increasingly analyzed quantitatively in biomedical research. Segmenting individual cell nucleus is an important step as many research studies involve counting cell nuclei and analysing their shape. We propose a novel weakly supervised instance segmentation method trained with image segmentation masks only. Our system comprises two models: an implicit shape Multi-Layer Perceptron (MLP) that learns the shape of the nuclei in canonical coordinates; and 2) an encoder that predicts the parameters of the affine transformation to deform the canonical shape into the correct location, scale, and orientation in the image. To further improve the performance of the model, we propose a loss that uses the total number of nuclei in an image as supervision. Our system is explainable, as the implicit shape MLP learns that the canonical shape of the cell nuclei is a circle, and interpretable as the output of the encoder are parameters of affine transformations. We obtain image segmentation performance close to DeepLabV3 and, additionally, obtain an F1-score$_{IoU=0.5}$ of 68.47% at the instance segmentation task, even though the system was trained with image segmentations.

**Keywords:** Instance Segmentation, Weakly-Supervised Learning, Implicit-Functions, Explainable

## 1. Introduction

Microscopy images are widely used in biomedical research to enable measurement of a variety of phenotypes in cells and whole organisms (Boutros et al., 2015). Automated analysis of biological images, and cell microscopy images in particular, increases the pace and quality of biomedical research (Caicedo et al., 2019; Mattiazzi Usaj et al., 2016). Increasing desire to analyze larger datasets and to measure more complex properties of living organisms call for the development of automated methods for biological image analysis.

In this paper, we address weakly-supervised instance segmentation of cell nuclei, or the pixel-wise segmentation of individual cell nuclei with incomplete ground-truth labels. Instance segmentation is the task of identifying the pixels that belong to a particular object class while, at the same time, distinguishing between different objects of interest. As opposed to binary segmentation where we are only able to identify the pixels in the image

displaying cell nuclei, instance segmentation allows us to count the number of cell nuclei present in the field of view, analyze their individual shapes and extract other scientifically informative features.

Existing tools for biological image analysis (McQuin et al., 2018; Schindelin et al., 2012) still require expertise in selecting and manually configuring the right algorithms for each image. Automated instance segmentation methods, especially in deep learning, also typically require pixel-wise annotations that distinguish each individual nucleus. These annotations are challenging to acquire, and ground truth at cell nuclei boundaries is not clearly defined (Segebarth et al., 2020). Our work directly addresses this problem by relaxing the labeling requirement while increasing explainability and interpretability of the model.

Existing fully automated Deep Learning methods typically compare favorably against expert researchers using hand-crafted biological image analysis tools to segment nuclei in microscopy images (Caicedo et al., 2019). Popular deep networks for bio-image semantic segmentation include the U-Net (Ronneberger et al., 2015; Vuola et al., 2019) and DeepLabV3 architectures (Chen et al., 2018, 2017). While these popular architectures have attained state-of-the-art performance in many semantic segmentation tasks, they are not explicitly designed for instance segmentation. For instance segmentation, the Cellpose model (Stringer et al., 2021) incorporates a U-Net into a gradient vector model to segment individual cell instances. Another deep network architecture well suited to cell nuclei instance segmentation model is Mask R-CNN (He et al., 2017). These models are trained in fully supervised fashion and require ground truth segmentation masks of individual cells.

Weakly supervised instance segmentation models typically exploit the availability of ground truth bounding box annotations to develop a model capable of predicting a segmentation mask for each instance in the image (Khoreva et al., 2017; Zhou et al., 2018). We follow a different approach, where we use binary segmentation masks instead of bounding boxes, but offer increased explainability and interpretability.

Other weakly supervised instance segmentation models use single point annotations to train (Tian et al., 2020; Qu et al., 2020; Nishimura et al., 2021). These annotations are faster to obtain than bounding box or segmentation masks, and the authors show that they are able to segment cell nuclei with a small performance loss when compared with a fully supervised segmentation model. However, these approaches are still not explainable nor interpretable, which are properties that could benefit researchers and pathologists.

We propose to let a model learn an explainable canonical shape representation of the instances in the dataset. Then, the task of instance segmentation is reduced to the deformation of the learned canonical shape, using an affine transformation, for each instance in the image. Our method consists of 2 models: 1) an implicit shape Multi-Layer Perceptron (MLP) model that learns a single canonical shape of the cell nuclei; and 2) an encoder that predicts the location, orientation, and size of each nuclei in the image. Our encoder follows a similar architecture to DeepLabV3 and we obtain comparable results to DeepLabV3 in terms of image segmentation, with the added benefit of performing instance segmentation.

**Contributions:** We propose a novel Weakly Supervised model capable of performing instance segmentation using image segmentation as the supervisory signal. Our model is explainable, capable of summarizing the shape of the instances present in the dataset. Additionally, our encoder is interpretable, predicting parameters (translation, scale, and rotation) that are understandable.

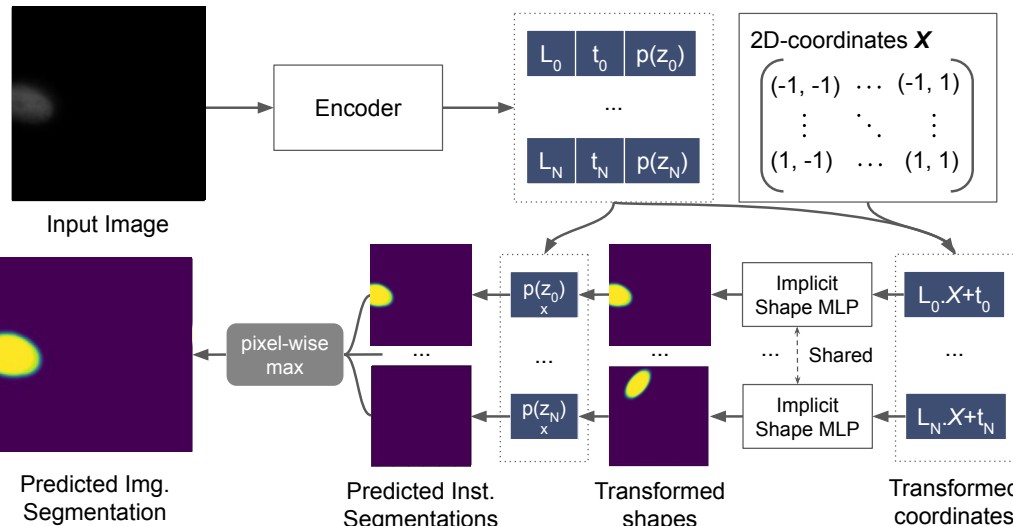

Figure 1: Our system performs instance segmentation while trained on binary segmentation masks and is composed by 2 models: an encoder and a canonical shape MLP. The encoder predicts the deformation $L$, location $t$ and presence $p(z)$ of $N$ instances. These parameters are used to transform a grid of 2D pixel coordinates $X$ before being given to a canonical shape MLP that predicts $N$ instance segmentation masks. These masks are combined using the maximum operator into an image binary segmentation mask.

## 2. Proposed Method

We propose to simultaneously learn the canonical shape of the objects to segment, and learn an affine transformation to place $N$ instances in the correct location, orientation, and scale. The canonical shape of the object is learned with a MLP that predicts the likelihood of a 2D cartesian coordinate belonging to the canonical shape. Then, an encoder predicts the parameters of $N$ instances in the image that are used to deform and place the instances in the correct location, scale, and orientation. Finally, the $N$ instances are combined to form a binary segmentation prediction for an image. An outline of our method is shown in Figure 1.

### 2.1. Implicit Shape Representation

We propose to represent the canonical shape of the objects of interest by modeling each $(x, y)$ pixel coordinate of a predefined grid $X$ as a bernoulli distribution. The distribution is parameterized using an implicit shape MLP that predicts $p(s|x, y, r)$, where $r$ is the distance to the origin $r = \sqrt{x^2 + y^2}$, and $s$ is a bernoulli random variable representing the value of the canonical shape at location $(x, y)$.

Note that the likelihood of $s$ only depends on $(x, y)$ and is independent of input image intensities. Also, as $(x, y)$ are continuous values, we can sample the desired shape at different

resolutions and in different positions. This is an important feature as the model should be able to segment objects of different scales and positioned in any location of the image.

The implicit shape MLP consists of three layers, with 8, 4, and 1 output channels respectively, followed by a ReLU activation function, except for the output layer which is followed by a negative Softplus activation function to model the log likelihood of $s$. The input $(x, y, r)$ is concatenated with the output of the first two layers, to make that information available at each level of the model. The implicit shape MLP is trained jointly with the encoder as described in section 2.2.

To compose the canonical shape, we sample a grid of $(x, y)$ pixel coordinate values $X$ with the desired image resolution, between fixed values for $x$ and $y$ (*i.e.* $-1 \leq x \leq 1$ and $-1 \leq y \leq 1$). To change the scale, orientation, and location of the canonical shape in the image, we transform $X$. We apply an affine transformation $T = [L|t]$ to $X$, where $L$ is a $2 \times 2$ lower triangular matrix and $t$ is a 2D vector. Therefore, the transformed grid $X'$ becomes:

$$X' = L.X + t. \tag{1}$$

We constraint the diagonal of $L$ to be positive but allow all the remaining degrees of freedom of $T$ to take any real value. Then, we compute $p(s|x', y', r')$ for each value of the grid to obtain an image of the transformed canonical shape.

## 2.2. Image Segmentation

With the canonical shape and the possibility to scale, rotate, and translate the shape, it becomes possible to segment individual objects. The task of segmenting an object instance in an image becomes to predict the parameters of the transformation $T$.

We train an encoder to predict $N$ affine transformations $T_i$ and an additional parameter $p(z_i)$ to indicate if the given instance is present in the image or not. As we train the model with a single ground-truth binary segmentation mask, we need to combine the $N$ instance masks into a single image segmentation mask. We define the likelihood of a single 2D pixel coordinate as the maximum likelihood over all $N$ predicted instances:

$$p(y|I) = \max_{1 \leq i \leq N} p(s_i|x_i', y_i', r_i')p(z_i), \tag{2}$$

where $I$ is the input image and $p(y|I)$ is the segmentation prediction for a single pixel. After computing $p(y|I)$ for all pixels in the input image, we end up with a binary segmentation mask, as can be seen in Figure 1. Then, any standard binary segmentation loss can be applied. In this work, we simultaneously optimize the binary cross entropy loss and the Dice loss. This architecture can be trained end-to-end, meaning that both the Implicit Shape MLP and the Encoder are simultaneously trained by minimizing the segmentation loss.

## 2.3. Regularization and Postprocessing

By training the model to minimize an image segmentation loss, there is no guarantee that it will be good at segmenting individual instances. If the number of predicted instances $N$ is much larger than the actual number of instances $K$ in an image, the model may compose

a single cell nucleus by multiple individual instances, to accommodate for small deviations from the canonical shape.

However, it is possible to include additional supervision on top of the predicted parameters to improve the instance segmentation performance, without providing instance segmentation masks. We apply a L2 loss on the number of predicted instances $\sum_{i}^{N} p(z_i)$ to be close to the ground-truth number of instances $K$ in the input image:

$$\mathcal{L}_{cnt} = \left[ \left( \sum_{i}^{N} p(z_i) \right) - K \right]^2 . \tag{3}$$

Even though the $\mathcal{L}_{cnt}$ loss promotes the prediction of the correct number of instances in an image, there may still occur some situations where the same instance is represented by multiple predictions. In order to improve the instance prediction performance, we group together one bigger instance $b$ with a smaller instance $m$ if $m$ is contained in $b$. To identify if $m$ is contained in $b$, we compute the ratio between their intersection with the area of $m$. Then, we group $b$ and $m$ if $\frac{b \cap m}{m} \geq 0.5$.

### 2.4. Model Architecture

We use ideas from state-of-the-art image segmentation models to design our model. Our model is composed of a Resnet50 (He et al., 2016) backbone, without the Global Average Pooling and final Fully-Connected layers used for the classification. Additionally, we replace the last strided convolutional layer with a dilated convolution, to avoid downsampling the feature map resolution without losing contextual information. We use $128 \times 128$px resolution images, and the extracted features have a resolution of $8 \times 8$. Finally, in a similar fashion to what standard one-stage object detection architectures do, we add two $1 \times 1$ convolutional layers to predict the parameters of the transformation $T$ and $p(z)$ of 2 instances. This results in $N = 2 \times 8 \times 8 = 128$, for $128 \times 128$px resolution images.

We use the predicted parameters to transform the grids $X$ with a resolution $4\times$ lower than the original image. After combining the $N$ instance segmentations using Equation 2, we upsample the resulting image segmentation mask to the same resolution as the input image. During inference we can use grids $X$ of the same resolution as the input images for improved instance segmentation resolution.

We train the model to minimize $\mathcal{L}_{BCE} + \mathcal{L}_{Dice} + \alpha \mathcal{L}_{cnt}$, where $\mathcal{L}_{BCE}$ is the Binary Cross-Entropy Loss, $\mathcal{L}_{Dice}$ is the smooth Dice Loss and $\alpha$ was set to $1e^{-2}$ after hyperparameter tuning. The model was trained with Stochastic Gradient Descent with a nesterov momentum of 0.9. The learning rate was scheduled using the One Cycle learning rate policy (Smith and Topin, 2019), with a maximum learning rate of $1e^{-2}$. The model was trained for 50 epochs.

## 3. Evaluation

### 3.1. Dataset

To evaluate our approach, we used a dataset from the Broad Bioimage Benchmark Collection with accession number BBBC038 (Caicedo et al., 2019). This dataset was part of the 2018

Table 1: We evaluate our model with different resolutions for the grid $X$, the effects of our postprocessing method (+P) and of our loss on the number of instances $\mathcal{L}_{cnt}$.

| | | | Image | | Instance | |
|---|---|---|---|---|---|---|
| Approach | Grid Res. | Unsup. IS | IoU | Dice | $F1_{IoU=0.5}$ | $F1_{mean}$ |
| DeepLabV3 | - | | 74.29 | 85.25 | - | - |
| Mask R-CNN | - | | 82.15 | 90.20 | 78.75 | 69.81 |
| Ours | $32 \times 32$px | ✓ | 71.38 | 83.30 | 32.49 | 34.08 |
| Ours | $128 \times 128$px | ✓ | 72.25 | 83.89 | 30.55 | 32.39 |
| Ours+P | $128 \times 128$px | ✓ | 72.25 | 83.89 | 57.57 | 51.99 |
| Ours+$\mathcal{L}_{cnt}$ | $32 \times 32$px | ✓ | 70.60 | 82.77 | 65.10 | 56.43 |
| Ours+$\mathcal{L}_{cnt}$ | $128 \times 128$px | ✓ | 71.02 | 83.06 | 65.36 | 57.00 |
| Ours+$\mathcal{L}_{cnt}$+P | $128 \times 128$px | ✓ | 71.02 | 83.06 | 68.47 | 60.27 |

Data Science Bowl challenge in Kaggle and contains a total of 841 images from 31 different biological experiments. From these images, 670 were used to train the model, 65 images were used for validation and 106 for testing.

The dataset contains 2D light microscopy images of stained nuclei, including fluorescent and tissue images. Fluorescent images display cells of different sizes and are stained with DAPI and Hoechst. Tissue images belong to different organs and animal models and are stained with hematoxylin and eosin. The images were annotated by experts with a segmentation mask for each individual nucleus. There is no overlap between different nuclei masks, meaning that a pixel in the input image may only be assigned to a single nucleus.

Since the images are of varying resolution, we break them into patches of a fixed resolution of $128 \times 128$ pixels (px). Additionally, we sum the individual segmentation masks into a single binary patch segmentation mask that we use as ground-truth to train the model. The number of instances $K$ is obtained by identifying the number of instance segmentation masks with at least one pixel segmented.

## 3.2. Image Segmentation

We compared the performance of our model with DeepLabV3 for a reference in image segmentation, as the architecture of our encoder is similar to DeepLabV3. We used a pretrained DeepLabV3 model and fine-tuned it on the BBBC038 dataset in the same conditions as our model.

We performed ablation studies on the $\mathcal{L}_{cnt}$ loss, the size of the sampling grid $X$ during inference and the effects of the proposed post-processing method. Regarding the resolution of the sampling grid, we tested with 2 different resolutions: 1) $32 \times 32$px, the same resolution used during training; and 2) $128 \times 128$px, the same resolution as the input image. The results can be seen in Table 1.

Even though the model was trained to produce $32 \times 32$px instance segmentation masks that are then upsampled, the model gets slightly better performance if the instance segmentation masks are sampled from the implicit shape MLP at the original image resolution.

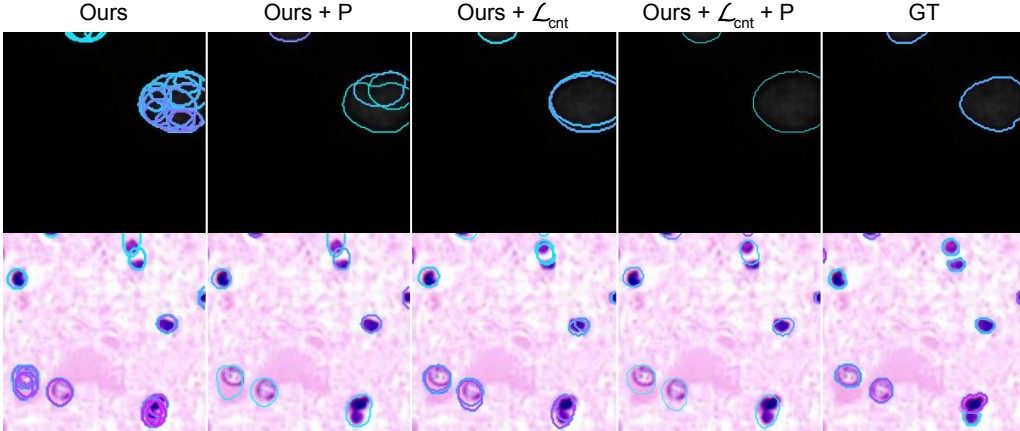

Figure 2: Instance segmentation results of the different proposed approaches. The baseline predicts many instances for each GT instance, leading to poor instance segmentation performance. The addition of the $\mathcal{L}_{cnt}$ loss improves the results by reducing the number of predicted instances. The postprocessing (P) method further refines the results by merging instances with large overlap.

Our model obtains an IoU and Dice slightly bellow the DeepLabV3 baseline. Additionally, when we apply a loss on the number of predicted objects $\mathcal{L}_{cnt}$, the image segmentation performance slightly decreases. This could be due the fact that, when an object is not a perfect circle or of elliptical shape, the model without $\mathcal{L}_{cnt}$ composes the irregular object with multiple small predicted instances, as shown in the first column of Figure 2. By penalizing the usage of larger number of predicted instances for each image than the actual number of objects, the model is forced to compromise and incur in some segmentation error in those irregular instances.

The image segmentation results are close to those of DeepLabV3, that has a similar architecture to our encoder, but it performs worse than Mask R-CNN. This may indicate that

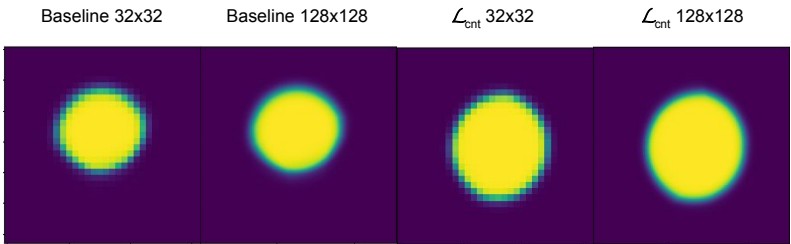

Figure 3: The canonical shape learned by the baseline and the $\mathcal{L}_{cnt}$ models when sampled at different resolutions. Even though most of the instances in the dataset are elliptical, the model learned a circular canonical shape.

the Mask R-CNN architecture is better suited for this particular task than the DeepLabV3 model and that the image segmentation performance of our model may be limited by the encoder architecture.

### 3.3. Instance Segmentation

We evaluate the model at the task of instance segmentation. Notice that the model was not trained to perform the task of instance segmentation, the only supervision provided to the model was the binary image segmentation mask and, when we use $\mathcal{L}_{cnt}$, the number of individual instances in each patch.

In order to evaluate the instance segmentation performance, we compute the F1-score at specific IoU thresholds between the predicted and the GT instances. More specifically, we test the F1-scores with IoU thresholds starting at 0.1 up to 1.0 with 0.1 intervals. We report the F1-score at an IoU of 0.5, and the mean of the F1-scores at all tested IoU thresholds.

In Table 1, we can see that the addition of $\mathcal{L}_{cnt}$ drastically improves the instance segmentation performance over our baseline. Our postprocessing method (+P) improves the instance segmentation performance of our baseline and of the model trained with $\mathcal{L}_{cnt}$, even though the improvements in the $\mathcal{L}_{cnt}$ version are more modest. As the baseline produces many overlaping instance predictions, as seen in Figure 2, the postprocessing method merges these predictions into more bigger shapes closer to the GT instances. On the other hand, the $\mathcal{L}_{cnt}$ model predicts a smaller number of instances and with shapes closer to the GT.

It is possible to inspect the canonical shape of the nucleus that the model learned by supplying the canonical grid $X$ to the implicit shape MLP. The resulting canonical shapes are shown in Figure 3. We can see that the model learned to predict a circle as the canonical shape, even though most of the objects in the dataset are elliptical. However, this result is intuitive as it is easy to deform a circle into ellipses and, therefore, we argue that is a good description of the shapes of the dataset.

## 4. Conclusion

We present a method that simultaneously learns the shape representation of the objects in the BBBC038 dataset and performs instance segmentation. Additionally, the model is trained with image segmentation masks and not with instance segmentation masks. We show improvements to instance segmentation performance either by adding a loss on the number of predicted instances or by postprocessing the results.

In the future we want to allow our implicit shape MLP to learn multiple shapes instead of just one. This feature is important in datasets with objects of different shapes, and even to accommodate for variations and hierarchical structure not captured by our deformation model.

This architecture also opens the door to the inclusion of domain knowledge in the training of segmentation models. It may be possible to apply additional supervision to the output of the encoder, for instance, to impose a mean size (scale) of cell nuclei. It may also be possible to apply domain knowledge to the implicit shape MLP, by regularizing or applying supervision to the learned canonical shape.

## Acknowledgments

The project TAMI - Transparent Artificial Medical Intelligence (NORTE-01-0247-FEDER-045905) partially funding this work is co-financed by ERDF - European Regional Fund through the North Portugal Regional Operational Program - NORTE 2020 and by the Portuguese Foundation for Science and Technology - FCT under the CMU - Portugal International Partnership.

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
