# OpenReview forum: "Explainable Weakly-Supervised Cell Nuclei Segmentation by Canonical Shape Learning and Transformation"
_MIDL.io/2022/Conference — MIDL 2022_

### Official Review · Reviewer_nZoD · 2022-01-19

**Confidence:** 4
**Preliminary Rating:** 2
**Recommendation:** Poster

**Summary:**

This paper proposes a novel weakly supervised instance segmentation method for cell segmentation in microscopy images using canonical shape learning and transformation. By combining an implicit shape MLP with an encoder to learn affine transformation parameters the described approach fulfills explainability as well as interpretability constraints, which are especially important in medical use cases. Furthermore, due to its weakly-supervised nature, the ever-accruing issue with labor intensive and error prone data annotation is addressed.

**Strengths:**

In general, the contribute of the paper is well defined and it is designed and structured in a concise way. With the increasing complexity of nowadays DL systems and the coupled absence of explainabilty as well as interpretability, it is nice to see approaches trying to incorporate prior knowledge like shape information into the learning process. By solely relying on binary segmentation masks the need for instance annotation can be circumvented and may reduce the overall number of human resources needed to generate ground truth data sets.
A further positive aspect is the conduction of ablation studies to show the influence of the different algorithmic steps.


**Weaknesses:**

My major concern with this work is the overall benefit of the proposed approach. While relying on weakly supervision, explainability and transferability has its merits, in the end the segmentation performance is the key value that counts. This is especially critical in the healthcare domain. Biologist and doctors won't rely on systems giving them subpar performance, if with admittedly more time-consuming annotation work more powerful and accurate methods can be applied. To tackle annotation effort, nowadays a vast amount of public expert annotated data sets is available, especially in the cell segmentation domain (e.g. [1-3]), which can be used to pretrain models or in Transfer Learning approaches.
Another concern is the limited use case. The utilized dataset does not contain different magnifications, different complexities of cell shapes etc. and therefore the generalization ability of this approach is questionable. Although when deliberately looking at segmenting cell nuclei, circular shapes seem appropriate, but in general other non-circular shaped cells/nuclei exist.
To incorporate the Lcnt loss, the number of cells in the ground truth images is needed. It would be interesting to see the actual time difference in creating binary segmentation mask and cell counts compared to creating instance segmentation masks. This could be a key performance value to argue about the importance of the proposed approach.
The overall idea, as already stated above is very interesting, as it allows to incorporate prior expert knowledge. Is there a reason for the selection of affine transformations and not making the additional step into nonrigid registration? This could theoretically enhance the approach to be able to cope with non-circular shapes and thereby hopefully also further increase the segmentation accuracy.
As described by the authors the role of the encoder architecture is not further explored but seems very important for the final performance. This could additionally be interesting in terms of needed computational resources, if models with a lower number of parameters could possibly also lead to good results.
Is there any improvement in computational costs, e.g in inference time or flops? On par or slightly worse performance but with better computational profile is favorable in terms of sustainability and speed. This could be an argument for the proposed approach, if applicable.
The post processing step is critical, as it will also merge overlapping cells and is strongly dependent on the selected threshold. What happens if the smaller instance is actually correct and the bigger one is wrong? A possible example can be seen in the lower right of Figure 3.
Is there a reason for not comparing the proposed approach to other weakly supervised segmentation algorithms?

[1] Kromp, F., Bozsaky, E., Rifatbegovic, F. et al. An annotated fluorescence image dataset for training nuclear segmentation methods. Sci Data 7, 262 (2020). https://doi.org/10.1038/s41597-020-00608-w
[2] Vrabac, D., Smit, A., Rojansky, R. et al. DLBCL-Morph: Morphological features computed using deep learning for an annotated digital DLBCL image set. Sci Data 8, 135 (2021). https://doi.org/10.1038/s41597-021-00915-w
[3] Hou, L., Gupta, R., Van Arnam, J.S. et al. Dataset of segmented nuclei in hematoxylin and eosin stained histopathology images of ten cancer types. Sci Data 7, 185 (2020). https://doi.org/10.1038/s41597-020-0528-1


**Deanonymize Review:**

no

**Detailed Comments:**

•	Especially for weakly supervised instance segmentation a more up to date state of the art would be appreciated, e.g. [4].
•	Table 1: I assume with +P you reference to your postprocessing, not pre-processing steps.
•	Table 1: For readability highlighting the best performing approaches would be helpful.
•	Figure 3 would benefit from a more detailed description in the caption: P, Lcnt, Baseline?

[4] Kazuya Nishimura, Chenyang Wang, Kazuhide Watanabe, Dai Fei Elmer Ker, Ryoma Bise, Weakly supervised cell instance segmentation under various conditions, Medical Image Analysis, 73, 2021, https://doi.org/10.1016/j.media.2021.102182


**Final Rating After The Rebuttal:**

3: Borderline

**Justification Of The Final Rating:**

I want to thank the authors for addressing my concerns. Although I understand the focus lying on architecure design and interpretability/explainability, I'm still sceptical about the applicability of the approach in a real setting. A more thourogh evaluation with more complex data sets and a in depth comparision to weakly supervised methods would neede to be finally convinced.

**Paper Type:**

methodological development

**Questions To Address In The Rebuttal:**

It would be important to emphasize and show the added value of the proposed weakly supervised approach in terms of computational efficiency, time saving concerning annotations and generalization ability. In other words, convincing facts that justify the below state of the art segmentation performance.

**Special Issue:**

no

---

### Official Review · Reviewer_cH4Y · 2022-01-24

**Confidence:** 4
**Preliminary Rating:** 4
**Recommendation:** Oral, Poster

**Summary:**

This work proposes a new instance segmentation method for cells in microscopy images.  The method is weakly supervised in the sense that only binary segmentations are used during training, instead of instance segmentations.  The key idea is the combination of two models, (i) a MLP that learns a canonical representation of the shape of the object of interest, and (ii) an encoder that estimates affine parameters to transform the canonical representation to the correct position and shape in the image. Different modifications are proposed: a post processing step to merge overlapping instances and a loss term to restrict the number of instances. Experiments are conducted to compare to two semantic and instance segmentation models and an ablation study is performed.

**Strengths:**

- The authors present a novel approach to perform instance segmentation using only binary image segmentation masks. The paper is well motivated and well written.

- The main novelty lies in learning a canonical shape of the object of interest and finding the position and scale of this shape for each instance.

- Variation of the proposed methods include a post-processing step to reduce the number of predicted instances and an additional loss term to restrict the number of instances directly in the training. The related experiments are convincing.


**Weaknesses:**

- The work only compares to two baseline methods: DeepLabV3 for binary image segmentation and Mask R-CNN both for instance segmentation. The proposed method performs worse compared to both approaches. The authors argue that

> this may indicate that the Mask R-CNN architecture is better suited for this particular task than the DeepLabV3 model and that the image segmentation performance of our model may be limited by the encoder architecture.”

&ensp; &ensp; This is merely a speculation.

- So far, a simple shape of a circle/ellipse is learned. It remains unclear if more complicated shapes are possible.

- Details on the training procedure is missing. Some open questions are: How exactly are the two models combined? How exactly is the loss function computed? Are the shape representations used as a mask, both on image and GT? What is the initial shape? How to choose N, or is it done automatically?

- The captions of figures 3 and 4 need more detailed captions.

**Deanonymize Review:**

no

**Detailed Comments:**

- The name ‘baseline’ in  Figs 3 and 4 is misleading. The authors mean here the proposed approach without post-processing or additional loss term. But the name ‘baseline’ is misleading, since it often refers to state-of-the-art comparison method (here DeepLabV3 and Mask R-CNN).
Please provide more detailed captions for both figures.

- Why did the authors restrict the transformation of the canonical shape to linear (affine) transformation? Isn’t this too restrictive? For more complicated shapes, an elastic deformation model would be more suitable.

- Some literature review on weakly supervised learning in the context of semantic/instance segmentation is missing.

- A simple shape of a circle/ellipse is learned. Are more complicated shapes possible? A discussion on this is needed.

**Final Rating After The Rebuttal:**

4: Weak Accept

**Justification Of The Final Rating:**

I acknowledge the authors effort to address the reviewer's concerns. I keep my initial recommendation of a weak accept, still having concerns about the extention to other shapes and deformation models.

**Paper Type:**

methodological development

**Questions To Address In The Rebuttal:**

- Please add more detailed figure captions.

- Clarify the open questions regarding the training procedure (see above).

- A clarification/discussion is needed why the proposed model performs so badly compared to the two baseline methods both for image and instance segmentation.

**Special Issue:**

yes

---

### Official Review · Reviewer_DCBu · 2022-01-24

**Confidence:** 4
**Preliminary Rating:** 2

**Summary:**

This paper proposes a novel method for nuclei instance segmentation with binary image segmentation masks only. The key idea is to adopt an implicit shape representation for nuclei instances, and thus to learn instance representations without instance supervision. The proposed pipeline consists of two main modules: (1) a MLP to learn an implicit nuclei shape representation in canonical coordinates, and (2) a encoder to predict affine transformation parameters to deform canonical shapes to align with the nuclei instances in the image. To show the efficacy of the proposed method, the authors conduct experiments on one benchmark. The key idea of this paper is interesting and reasonable, while the experimental comparisons are far from sufficient.

**Strengths:**

* The paper is mostly well-written and easy to follow.
* The key idea, which is to adopt an implicit shape representation for nuclei instances and thus to learn instance representations without instance supervision, is interesting and reasonable.
* The ablation results show the efficacy of the module design of the proposed method.

**Weaknesses:**

* The title says this paper is about cell segmentation, while "cell" and "nuclei" are both used in the paper. It is not clear which target this paper is focused on. The experiments are conducted on a nuclei segmentation dataset only.
* The proposed method requires number of instance in each image as annotations, which can have similar cost to point annotations for each nucleus. So there should be comparisons to other weakly-supervised nuclei segmentation papers using point annotations, e.g.,

[1] Qu, Hui, et al. "Weakly supervised deep nuclei segmentation using points annotation in histopathology images." International Conference on Medical Imaging with Deep Learning. PMLR, 2019.

[2] Tian, Kuan, et al. "Weakly-Supervised Nucleus Segmentation Based on Point Annotations: A Coarse-to-Fine Self-Stimulated Learning Strategy." International Conference on Medical Image Computing and Computer-Assisted Intervention. Springer, Cham, 2020.

(Note that the proposed method still require mask annotations while these two papers only need points.)

* Can the proposed method handle cell/nuclei instances with large shape variety? How does this method perform on nuclei datasets such as MoNuSeg (Kumar, Neeraj, et al. "A dataset and a technique for generalized nuclear segmentation for computational pathology." IEEE transactions on medical imaging 36.7 (2017): 1550-1560.) ?
* The experimental results are only conducted on one benchmark, which can be potentially biased.
* In Table 1, DeepLabV3 can have instance predictions by applying simple peak detection post-processing, while the results are not shown.
* How is the computation cost of each method in Table 1? The training and inference time can be discussed.

**Deanonymize Review:**

no

**Detailed Comments:**

* What is the range of N in equation (3), the number of predicted instances? Is it fixed for all images? Is there any evaluation of the accuracy of the predicted number of instances?

**Final Rating After The Rebuttal:**

3: Borderline

**Justification Of The Final Rating:**

I do agree with the authors and appreciate their effort in promoting the "interpretability of the model’s predictions and the explainability of the canonical shape". Nevertheless, I still have concerns on whether their proposed method actually work in real situations, given only results on one benchmark with simple nuclei shapes. Also, it would make this paper stronger if their model in their proposed "weak" setting with annotations of binary masks and number of nuclei is compared numerically to other "weak" methods with point annotations.

**Paper Type:**

methodological development

**Questions To Address In The Rebuttal:**

The concerns discussed in weaknesses need to be addressed, especially those about discussion and experimental comparisons with other weakly-supervised methods, which should be conducted on more than one dataset.

**Special Issue:**

no

---

### Official Review · Reviewer_mwRe · 2022-01-25

**Confidence:** 4
**Preliminary Rating:** 5
**Recommendation:** Oral, Poster

**Summary:**

The authors propose an instance segmentation approach for cells by learning a canonical shape; for each cell, this shape is deformed via an affine transformation into the correct position and shape. Furthermore, the required labeling information to train such a model only contains the binary segmentation map and the number of instances.

**Strengths:**

The authors propose a novel idea to tackle instance segmentation tasks: the canonical shape and transformation information may lead to better explainability and provide information about cell morphology. Moreover, no perfect instance segmentation labels may be needed since, for training an image, only the binary segmentation label and the number of instances are required.

For certain circumstances, this paper may really present a viable approach, e.g., if cell morphology information is of interest or if not enough resources are available to process the expensive labeling for instance segmentation.
Overall, the paper is well-crafted. Nevertheless, the authors should ensure that the details of the model are clearly understandable.


**Weaknesses:**

The abstract's statement that only image segmentation masks are needed seems misleading: there has to be some way to extract the cell number for an input image. The authors should alter this statement accordingly. Furthermore, a discussion of how to efficiently acquire the number of cell instances would be insightful – especially if random crops are used for training and the number of instances changes.

Only a relatively simple dataset is used to test the model. It would be interesting to see how the model would deal with more intricate cell shapes. Furthermore, the authors should discuss how the approach responds to overlapping cells, artifacts, and other error sources.

It is not clear whether the transformed grid is input to the MLP (as described in Figure 2) or if the canonical shape is transformed afterward (Section "Transformation of Canonical Shape").
In addition, the authors should try to make Figure 2 more intelligible: e.g., the Figure should clearly state that 2D coordinates are also an input, or different colors could indicate that L and T are different from z.


**Deanonymize Review:**

no

**Detailed Comments:**

* What is the difference between z and p(z)?
* What exactly is the variable "s"?
* Some sentences don't have an oxford comma ("scale, rotate, and translate").

**Final Rating After The Rebuttal:**

5: Strong Accept

**Justification Of The Final Rating:**

The authors did address all of my concerns and questions. I still think this is an interesting approach worth sharing at this conference and I'm looking forward to the fully updated version of the paper and future work with the proposed method.

**Paper Type:**

methodological development

**Questions To Address In The Rebuttal:**

* How can this model be used to reduce labeling costs? Is there a way to efficiently obtain a binary segmentation and the number of instances? What happens if the images are cropped?

* How does the approach deal with overlapping cells?

**Special Issue:**

no

---

### Meta-Review · Area_Chair_fu6e · 2022-02-13

**Recommendation:** Accept (Poster)
**Confidence:** 4

**Metareview:**

I thank all reviewers for their time and effort spent reviewing this paper and their engagement with the rebuttal process. I also thank the authors for their detailed rebuttal and changes made to the manuscript. On balance there is sufficient support for the paper to be accepted and I look forward to seeing it presented at MIDL. But I would encourage to authors to bear in mind the reviewers’ comments going forward, either for the camera-ready version of the paper (preferably) or otherwise for a future journal publication. In particular, the reviewers suggested that the paper would benefit from validation on a wider range of benchmark datasets (Reviewer DCBu) and extension to more complex deformation models (Reviewer cH4Y).

---

### Decision · Program_Chairs · 2022-02-28

Accept